



# Sample preservation and pre-treatment in stable isotope analysis: Implications for the study of aquatic food webs

Marc J. Silberberger[1], Katarzyna Koziorowska-Makuch[1], Karol Kuliński[1], Monika Kędra[1]

[1]Institute of Oceanology Polish Academy of Sciences, Powstańców Warszawy 55, 81-712 Sopot, Poland

*Correspondence to*: Marc J. Silberberger (marcs@iopan.pl)

**Abstract.** Stable isotope analysis has become one of the most widely used techniques in ecology. However, uncertainties about the effects of sample preservation and pre-treatment on the ecological interpretation of stable isotope data and especially on Bayesian stable isotope mixing models remain. Here, Bayesian mixing models were used to study how three different preservation methods (drying, freezing, formalin) and two pre-treatments (acidification, lipid removal) affect the
estimation of diet composition for two benthic invertebrate species (*Limecola balthica*, *Crangon crangon*). Furthermore, commonly used mathematical lipid normalization and formalin correction were applied to check if they improve the model results. Preservation effects were strong on model outcomes for frozen as well as formalin preserved *L. balthica* samples, but not for *C. crangon*. Pre-treatment effects varied with species and preservation method and neither lipid normalization nor mathematical formalin correction consistently resulted in improved model outcomes. Our analysis highlights that particularly
small changes in $\delta^{15}N$ introduced by different preservation and pre-treatments display a so far unrecognized source of error in stable isotope studies. We conclude that mathematical correction of stable isotopes data should be avoided for Bayesian mixing models and that previously unaddressed effects of sample preservation (especially those arising from preservation by freezing) have potentially biased our understanding of the utilization of organic matter in aquatic food webs.

## 1 Introduction

The development of stable isotope analysis (SIA) to study ratios of naturally occurring stable isotopes (SIs) of carbon ($\delta^{13}C$) and nitrogen ($\delta^{15}N$) has advanced the understanding of aquatic ecosystem functioning and in particular the knowledge about trophic interactions (Middelburg, 2014). Although SIA has been successfully applied in ecological studies for over three decades, many uncertainties about the effects of sample preservation and treatment still remain (Lau et al., 2012; Schlacher and Connolly, 2014).

In principal, SIA consists of the complete combustion of dehydrated and pulverized samples in an elemental analyser and transferring the gaseous products to $CO_2$ and $N_2$, which are subsequently detected in a mass spectrometer allowing for distinguishing between molecules containing $^{12}C$ and $^{13}C$, as well as $^{14}N$ and $^{15}N$. Ship-based sampling and sampling in remote locations, however, seldom allow for immediate sample preparation and analysis. Accordingly, samples must be preserved until the time of analysis. In case of organisms and sediments it is commonly done by freezing, since fluid





preservation in formalin and/or ethanol is known to affect SI ratios (Edwards et al., 2002). Such preservation effects are typically strong for $\delta^{13}C$ (reported $\Delta\delta^{13}C$ typically from -1 to -2‰), while they are small or insignificant for $\delta^{15}N$ (reported $\Delta\delta^{15}N$ typically from -0.5 to 0.5‰) (Lau et al., 2012; Rennie et al., 2012; Turner et al., 2015). Nonetheless, in an increasing number of studies formalin preserved organisms are measured for SIs, since it allows using community samples collected for other purposes or museum specimens (Chandra et al., 2005; Ozersky et al., 2012). Such studies typically use correction

factors to adjust measured SI ratios (e.g., Chandra et al. (2005): addition of 1.28‰ to observed $\delta^{13}C$), although consensus about the appropriate correction has not been reached as the effect may vary with preservation duration, sample treatment, formalin brand, and studied taxa (Edwards et al., 2002; Rennie et al., 2012). Due to the general belief that freezing does not affect SI concentrations (Bosley and Wainright, 1999), these correction factors have been established in most cases by comparison of formalin fixed and frozen samples without control for changes in SI ratios that are caused by freezing itself

(Edwards et al., 2002; Lau et al., 2012). Some studies, however, found that freezing can have an effect on SI ratios of similar magnitude as preservation in formalin (Feuchtmayr and Grey, 2003; Syväranta et al., 2011). Furthermore, the reported effects of freezing on $\delta^{13}C$ in these studies were opposing: according to Feuchtmayr & Grey (2003) freezing caused a depletion of $\delta^{13}C$ in zooplankton by approximately 1‰, while Syväranta (2011) found an enrichment of $\delta^{13}C$ in the Asiatic clam (*Corbicula fluminea*) by approximately 2‰. For formalin preservation, however, both studies report an enrichment in

$\delta^{13}C$ in comparison to dried samples.

Another consideration in SIA is the use of sample pre-treatments to remove carbonates and/or lipids. When studying food webs, scientists are typically interested in organic (and not inorganic) carbon as the $\delta^{13}C_{org}$ of an organism closely reflects the $\delta^{13}C_{org}$ of its food (McCutchan et al., 2003). Inorganic carbon structures (e.g. carbonate shells) are strongly enriched in $^{13}C$ and thus they have higher $\delta^{13}C$ in comparison to organic tissue and are routinely removed from samples (Schlacher and

Connolly, 2014). This is commonly achieved through treatment with hydrochloric acid (HCl). Such a treatment, however, has been reported to have variable effects on $\delta^{15}N$ and therefore analysing samples in duplicates for $\delta^{13}C$ (acidified) and $\delta^{15}N$ (non-acidified) is recommended (Schlacher and Connolly, 2014). However, since this increases the total number of analysed samples many studies that investigate aquatic food webs use acidified samples for both SIs (e.g., Fredriksen, 2003; Jankowska et al., 2018).

In contrast, the carbon in lipids originates from the diet of an animal. Nonetheless, chemical lipid removal is common when SIA is used to study aquatic food webs. The reasons behind this are the fast carbon turnover in lipids and their tendency to have lower $\delta^{13}C$ in comparison to carbohydrates and protein (Focken and Becker, 1998; Tieszen et al., 1983), which are likely to introduce a strong bias in food web studies when samples with variable lipid content are analysed. Consequently, chemical lipid removal is recommended when samples with presumably differing lipid content are being analysed (Pinnegar and Polunin, 1999). Since chemical lipid removal can affect $\delta^{15}N$ ratios (Søreide et al., 2006), this approach also requires the

preparation of duplicate samples. However, since such chemical lipid removal is time consuming and increases the total number of analysed samples, many authors choose to use a mathematical lipid normalization of $\delta^{13}C$ instead, based on the C/N ratio (McConnaughey and McRoy, 1979; Post et al., 2007).





While all above mentioned studies have addressed effects of sample preservation or pre-treatment separately, detailed knowledge about potential interaction effects of preservation and pre-treatment is still missing. Furthermore, studies on preservation or pre-treatment effects have been so far limited to quantifying changes in SI ratios and the development of mathematical corrections to adjust data that were acquired from differently treated samples. Knowledge about whether and how the introduced changes and recommended mathematical corrections alter the ecological interpretation of data remain scarce. Le Bourg et al. (2020) found that isotopic niche parameters of the sea star *Marthasterias glacialis* remained overall unchanged by various preservation methods or could be corrected for after formalin preservation. The outcomes of Bayesian mixing models, however, are strongly affected by mathematical lipid normalization (Arostegui et al., 2019). Although Arostegui et al. (2019) demonstrated that the choice of lipid normalization can affect the result of mixing models, they did not assess whether chemical lipid removal would result in similar model outcomes as lipid normalization. Knowledge about potential effects of different preservation and pre-treatment methods or mathematical correction for formalin preservation on mixing model outcomes is completely missing.

Therefore, the objectives of the present study were to: (i) quantify how sample preservation and pre-treatment affect carbon and nitrogen SI ratios, (ii) identify potential interaction effects between preservation and pre-treatment methods, (iii) study how preservation and pre-treatment affect the results of Bayesian mixing models, (iv) assess whether lipid normalization and mathematical formalin correction should be used to adjust data for the use in such models. To address these objectives, we chose the Gulf of Gdansk (southern Baltic Sea) as study area and two benthic invertebrate species (*Limecola balthica* and *Crangon crangon*) as study objects. We assumed that SI ratios are affected by all preservation and pre-treatment methods. We aimed to determine whether an ecological interpretation of data is possible and how the uncertainties of the used methods should be addressed.

## 2 Material and Methods

### 2.1 Study location and sample collection

The sampling site in the Gulf of Gdansk (Fig. 1) is characterized by sandy sediments, and year-round relatively stable bottom salinity around 7 (Witek et al., 2003). During times of high freshwater outflow from the close by Vistula river, however, the river plume can extend over the sampling site (Cyberska and Krzymiński, 1988). The vertical extension of the plume varies between 0.5 and 12 m (Grelowski and Wojewódzki, 1996). The bottom water temperature follows a typical seasonal pattern ranging from approximately 4°C in winter to 15°C in late summer. The onset of the spring phytoplankton bloom in the study region typically occurs between February and April, with the phytoplankton biomass peaking in April (Witek et al., 1997). Following the spring bloom, the primary production in the Gulf of Gdansk remains high until October (Witek et al., 1997). However, pelagic community respiration can exceed primary production for short periods in late summer, when water stratification is strong and water transparency is reduced (York et al., 2001). Such conditions have been reported to cause temporary oxygen depletion in bottom water of the Gulf of Gdansk (York et al., 2001).



Samples were collected from s/y *Oceania* on 24th April 2019 at the study location in the Gulf of Gdansk, southern Baltic Sea (54.40°N, 19.08°E, depth: 32 m; Fig. 1). Two dominant benthic species, *Limecola balthica* and *Crangon crangon*, were chosen as study objects. Both species have a wide distribution in European waters and play an important role in the southern Baltic Sea ecosystem. The bivalve, *L. balthica* is a primary consumer that has the ability to switch between suspension- and

deposit-feeding (Törnroos et al., 2015), while the shrimp *C. crangon* is a benthic predator (Oh et al., 2001). For each species enough individuals for 60 replicate samples (Fig. 2A) were collected by use of a benthic dredge and kept alive in filtered seawater for 24 h to allow for gut clearance. Altogether 60 individuals of *L. balthica* (size: 1-1.5 cm) and 120 individuals of *C. crangon* (length: 1-1.5 cm) were collected. For each species, the collected material was divided into 3 equal parts, which were subsequently preserved by oven-drying at 60°C on board, freezing at -80°C, and in seawater with 10% formalin

(approx. 4% formaldehyde) buffered with borax (Fig. 2A). *Limecola balthica* was removed from its shell prior to oven-drying and freezing. Intact individuals were, however, used for formalin preservation, since community samples and museum specimens are typically preserved in that way.

Furthermore, potential OM sources of the collected fauna were sampled. Twelve replicate samples of surface sediment (0-1 cm depth) and subsurface sediment (1-3 cm depth) were collected with box corer sampler (sSOM – surface sediment OM;

ssSOM – subsurface sediment OM). Half of the samples were dried at 60°C and the remainder frozen at -20°C (Fig. 2C). A Niskin bottle was used to collect water from 10 m depth. Water was then filtered through pre-combusted and pre-weighed Whatman GF/F filters to collect pelagic particulate OM (pPOM). Additionally, water samples were taken from Vistula River (54.25°N, 18.95°E, Fig. 1) on 6th May 2019 to obtain riverine particulate OM (rPOM) samples following the same filtration procedures. Half of pPOM and rPOM samples were dried at 60°C and the other parts frozen at -20°C (Fig. 2B).

**2.2 Sample preparation and analysis**

After 6 weeks of preservation, all frozen and formalin preserved samples were dehydrated. This was achieved by oven-drying at 60°C for pPOM and rPOM and freeze-drying for sediments and fauna samples. Organisms preserved in formalin were soaked in water for 6 h. The water was exchanged twice during this time. *Limecola balthica* was then removed from their shells prior to dehydration. All samples were then ground to a fine powder with mortar and pestle and kept in a

desiccator until further processing.

In the next step lipids were removed from half of each of 3 sets (different preservation methods) of fauna samples (Fig. 2A) by extraction in Folch solution (chloroform-methanol 2:1 (v/v)) for 24 hours (Søreide et al., 2006). Consecutively extractant was exchanged twice to ensure the majority of lipid was removed. Samples were then air dried under a fume hood at room temperature.

After weighing the samples into silver capsules, half of them were repeatedly (×4) treated with 2 M HCl to obtain maximum of combinations for different preservation and pre-treatments methods (Fig. 2). After each individual acid wash, samples were oven-dried at 60°C. Similarly, all pPOM and rPOM samples were subject to acidification. Samples for sediment OM were prepared in duplicates with and without acidification.



Samples were analysed for carbon, nitrogen as well as their SIs ($\delta^{13}C$ and $\delta^{15}N$) with an Elemental Analyzer Flash EA 1112
Series interfaced to an Isotopic Ratio Mass Spectrometer IRMS Delta V Advantage (Thermo Electron Corp., Germany).
Isotopic ratios, $\delta^{13}C$ and $\delta^{15}N$, were calculated using the laboratory working pure reference gases: $CO_2$ and $N_2$ calibrated
against IAEA standards (CO-8 and USGS40 for $\delta^{13}C$ and N-1 and USGS40 for $\delta^{15}N$). SIs measurements are reported in the
commonly used delta ($\delta$) notation in parts per thousand (‰) relative to the international standard Vienna Pee Dee Belemnite
and atmospheric air for carbon and nitrogen, respectively.

**2.3 Statistical analysis**

A two-way ANOVA was used to test for differences in isotopic ratios between OM sources (4 levels) and to identify
potential preservation effects (2 levels) on the OM sources. To identify differences among fauna samples, a factorial
ANOVA with four factors was used: (i) species identity (2 levels), (ii) preservation method (3 levels), (iii) lipid removal (2
levels), and (iv) carbonate removal (2 levels). Shapiro-Wilk test in combination with QQ-plots of the residuals and Levene's
test were used to test for normality and homogeneity of variance, respectively. Three OM samples ($1 \times$ rPOM$_{frozen}$; $1 \times$
pPOM$_{dried}$; $1 \times$ ssSOM$_{dried}$) were identified as extreme outliers that caused a violation of ANOVA assumptions for one of the
isotopes. These samples were excluded from the analysis. Post-hoc tests of significant factors and interactions of factors
were done by Pairwise Comparisons of Estimated Marginal Means with Bonferroni correction applied.

Bayesian mixing models were run in MixSIAR (Stock et al., 2018) to identify impacts of all combinations of preservation
and pre-treatment on the estimated contribution of different OM sources to the diet of the collected fauna. Individual models
were run per species-preservation combination, with pre-treatment as covariate (fixed factor with four levels, i.e. all
combinations of lipid removal and acidification; Fig. 2A). We included three distinct potential OM sources in the model:
rPOM, pPOM, and SOM (i.e. surface + subsurface). However, as our results showed, SOM $\delta^{15}N$ was significantly affected
by freezing, therefore we ran every model twice with two different source data sets: (i) source data set A with dried SOM
and (ii) source data set B with frozen SOM.

For $\delta^{15}N$ a trophic fractionation ($\Delta^{15}N$) of 3.4‰ (SD 1‰) was used for each trophic level (TL) (Post, 2002). A scaled trophic
fractionation ($\Delta^{13}C$) was used for $\delta^{13}C$, assuming a trophic enrichment of 4‰ (SD 1.3‰) for the first trophic transfer from
OM source to primary consumer (Hobson et al., 1995; Renaud et al., 2015) and of 0.4‰ (SD 1.3‰) thereafter (Post, 2002).
A scaled trophic fractionation for $\delta^{15}N$ was not chosen since the collected fauna represents lower TLs (i.e. TL$_{L.\ balthica} \approx 2$;
TL$_{C.\ crangon} \approx 3$ (Nordström et al., 2009)) for which a linear approach can be used (Hussey et al., 2014). Since the studied
species occupy different TLs (Nordström et al., 2009), we analysed them separately in MixSIAR with the following trophic
enrichment factors (TEF): *L. balthica*: TEF$_{\delta^{13}C}$ = 4‰±1.3; TEF$_{\delta^{15}N}$ = 3.4‰±1.0; *C. crangon*: TEF$_{\delta^{13}C}$ = 4.4‰±2.6;
TEF$_{\delta^{15}N}$ = 6.8‰±2.0. The TEF were calculated by summing of the trophic fractionation for the necessary trophic steps to
reach TL2 (for *L. balthica*) and TL3 (for *C. crangon*). All models were run on "long" setting (i.e. chain length = 300,000;
burn = 200,000; thin = 100; chains = 3) with an uninformative prior. Gelman-Rubin diagnostics and Geweke diagnostics
were used to verify if models have converged.



The δ¹³C values of all fauna samples with lipids intact were subjected to two separate mathematical lipid normalization methods (Fig. 2A). For lipid normalization according to Post et al. (2007) all δ¹³C values from samples with C:N > 3.5 were normalized by the following formula:

$$\delta^{13}C_{normalized} = \delta^{13}C_{untreated} + (0.99 \times C:N - 3.32),$$
(1)

For lipid normalization according to McConnaughey and McRoy (1979) all δ¹³C values from samples with C:N > 4 were subjected to:

$$\delta^{13}C_{normalized} = \delta^{13}C_{untreated} + 6 \times \left(-0.207 + \frac{3.9}{1+287 \times \left(1 + \frac{1}{0.246 \times C:N - 0.775}\right) \div 93}\right),$$
(2)

Accordingly, samples with C:N < 3.5 and C:N < 4 were not affected by normalization according to Post et al. (2007) and
McConnaughey and McRoy (1979), respectively. Two separate factorial ANOVAs (one for each normalization formula) with four factors were conducted: (i) species identity (2 levels), (ii) preservation method (3 levels), (iii) lipid approach (2 levels), and (iv) carbonate removal (2 levels). This allowed testing for differences in δ¹³C between lipid normalization and samples with chemically removed lipids, while also controlling for potential interaction effects with species identity, preservation method, and acidification.

Furthermore, δ¹³C of all formalin preserved samples was mathematically corrected by adding 1.28‰ to the original data according to Vander Zanden et al. (2003), which was previously used to study OM cycling in benthic food webs (e.g., Chandra et al., 2005). MixSIAR was then used to assess how the two lipid normalization formulas as well as the formalin correction affect mixing model results.

All statistical analyses were run in R version 3.6.2 (R Development Core Team, 2019), making use of emmeans (Lenth,
2020) and MixSIAR (Stock and Semmens, 2016) packages.

## 3 Results

### 3.1 Preservation and pre-treatment effects

#### 3.1.1 Organic matter sources

Allochthonous (rPOM) and autochthonous (pPOM, sSOM, ssSOM) OM sources were distinct from each other for both
studied isotopes. Two-way ANOVA confirmed a significant difference between OM sources for δ¹³C [$F_{(3, 37)}$ = 2062.89, p < 0.001] and δ¹⁵N [$F_{(3, 37)}$ = 412.28, p < 0.001]. The following pairwise comparisons identified a significant difference for all pairs of OM sources, except surface and subsurface sediments, for δ¹³C and δ¹⁵N (Fig. 3).

No effect of preservation method on δ¹³C of the OM sources was observed. For δ¹⁵N, however, a significant interaction effect between OM source and preservation method was identified by two way ANOVA [$F_{(3, 37)}$ = 6.92, p < 0.001].
Pairwise comparison revealed that δ¹⁵N of sSOM and ssSOM was significantly depleted in frozen samples in comparison to





samples that were dried (Fig. 3). Across all SOM samples, $\delta^{15}$N was on average 0.6‰ lower in frozen samples than in samples that were preserved by drying.

### 3.1.2 Fauna

Both studied isotopes showed a clear distinction between collected species (Fig. 4). Across all preservation and pre-

treatments *C. crangon* was on average enriched by 1.5‰ and 2.4‰ for $\delta^{13}$C and $\delta^{15}$N, respectively, in comparison to *L. balthica*. Overall, $\delta^{13}$C was more variable than $\delta^{15}$N. The total range of $\delta^{13}$C for each species for all treatments exceeded the difference between species and the $\delta^{13}$C range of the two species overlapped (Fig. 4), which was not the case for the total range of $\delta^{15}$N.

In addition to this species effect, a significant effect of preservation and both pre-treatments on $\delta^{13}$C was identified by

ANOVA (Table 1). Each of these effects was independent and no interaction effect was identified. Pairwise comparisons identified a significant depletion of $\delta^{13}$C after formalin preservation in comparison to drying (p.adj < 0.001) and freezing (p.adj < 0.001). No significant difference between $\delta^{13}$C dried and frozen samples was found (p.adj = 0.36).

For $\delta^{15}$N, ANOVA identified significant interaction effects between species identity and preservation method as well as between species identity and carbonate removal (Table 1). The effect of both interactions on individual $\delta^{15}$N measurements

were low. However, post-hoc testing showed that sample acidification significantly affected $\delta^{15}$N in both species (p.adj$_{L. balthica}$ < 0.001; p.adj$_{C. crangon}$ = 0.02), while no effect of preservation method was identified as significant. The direction of change after acidification was opposite for both species. Acidification caused an average enrichment of $\delta^{15}$N in *L. balthica* samples by 0.4‰, but a depletion (-0.3‰) in *C. crangon* samples.

### 3.2 Lipid normalization

All samples with lipids intact (i.e. samples not treated with Folch solution) had C:N > 3.5 (Table 2) and accordingly all samples were normalized by Post et al. (2007) formula. Most *C. crangon* samples (24 out of 30) and one *L. balthica* (out of 30) had a C:N ratio between 3.5 and 4.0 and accordingly lipid normalization according to McConnaughey & McRoy (1979) was applied only to the remaining 6 *C. crangon* and 29 *L. balthica* samples.

Factorial ANOVA showed that $\delta^{13}$C after lipid normalization differed significantly between the studied species, preservation

methods, and acidification for both normalization formulas (Table 1). Furthermore, neither formula achieved a non-significant result for normalized data vs. chemical lipid removal (i.e. lipid approach in Table 1). In addition to the independent treatment effects, both normalization formulas introduced a significant interaction effect between species identity and lipid approach. This interaction effect was the result of each normalization formula performing well for only one species. Lipid normalization according to Post et al. (2007) performed well for *C. crangon*, but poor for *L. balthica* (Fig. 5).

More accurate lipid normalization of *L. balthica* was achieved by the formula of McConnaughey & McRoy (1979), which in turn gave worse results for *C. crangon*.




### 3.3 Bayesian mixing models

### 3.3.1 Preservation and pre-treatment effects

Bayesian mixing models for *L. balthica* indicated a strong preservation effect on the model outcomes (Fig. 6A). For dried
fauna pPOM was identified as main carbon source across all pre-treatments. For frozen and formalin preserved samples, however, the models suggest a mixed diet of pPOM and SOM. The estimated contribution of SOM was slightly higher for formalin preservation than for freezing. This preservation effect was observed in model runs regardless whether source data A or B was used.

Furthermore, the preservation effect was modified by pre-treatment methods. Sample pre-treatment with Folch solution
caused an increased estimated contribution of pPOM to *L. balthica* diet across all preservation methods in comparison to samples with no pre-treatment (Fig. 6A). The difference among treated and untreated samples, however, increased from dried samples to frozen and formalin preserved samples. In addition, the direction of the HCl treatment effect differed between dried samples and the other preservation methods. HCl treatment caused a slight decrease of pPOM contribution in dried samples. The consequence of this was that Folch solution and HCl treatment effects (if both applied) cancelled each
other out in dried *L. balthica* samples and accordingly the model outcomes for both treatments and no pre-treatment were almost identical. For frozen and formalin preserved samples the HCl treatment caused an increased importance of pPOM (i.e. same direction as Folch solution). Consequently, model results for samples that received both pre-treatments differed from samples with no pre-treatment.

In contrast to *L. balthica*, model results for *C. crangon* indicated a strong reliance on SOM across all preservation methods
(Fig. 6A). The estimated contribution of pPOM was slightly higher for frozen and formalin preserved samples, but the difference was small. The used OM source data, however, had a considerable impact on the model outcome. Across all preservation and pre-treatments, source data set B (frozen SOM) resulted in an increased contribution of pPOM to *C. crangon* diet.

Sample pre-treatment with Folch solution caused an increased estimated contribution of pPOM (Fig. 6A). The difference for
this increase was low among preservation methods but increased when source data set B was used in the model. HCl treatment caused only minor changes to model outcomes. For samples that received both pre-treatments, however, HCl treatment seems to cancel out the effect of prior treatment with Folch solution.

### 3.3.2 Mathematically corrected data

Formalin correction resulted in a changed model outcome for *L. balthica*. Across all pre-treatments the estimated
contribution of pPOM increased in comparison to uncorrected formalin preserved samples (Fig. 6A). However, the change for samples without a pre-treatment was weakest. All samples that received any pre-treatment were estimated to rely almost exclusively on pPOM, similar to dried samples. Samples that received no pre-treatment were estimated to rely equally on pPOM and SOM.





Formalin correction also lead to an increased contribution of pPOM to the diet of *C. crangon* (Fig. 6A). Since the reliance of
*C. crangon* on pPOM was low for all preservation methods, results after formalin correction differed from all preservation methods.

The effects of mathematical lipid normalization were quite variable (Fig. 6B). In contrast to the above described species-specific applicability of the two normalization methods, modelling results for both methods were similar. Only when lipid normalization was applied to formalin preserved samples, modelling results clearly differed between the two methods.
Modelling results of lipid normalized data generally failed to resemble results for samples that were treated with Folch solution (Fig. 6B). Lipid normalization had virtually no impact on modelling results for *C. crangon* and results for lipid normalized data mirrored the results of the original, and not normalized, data.

The lipid normalization effects on *L. balthica* models were more variable. Both normalization methods successfully altered model outcomes for dried samples (Fig. 6B). For frozen samples, however, the estimated contribution of pPOM was
overestimated after lipid normalization in comparison to lipid extracted frozen samples. For formalin preserved *L. balthica* samples lipid normalization according to McConnaughey and McRoy (1979) successfully corrected modelling outcomes. Normalization according to Post et al. (2007), however, overestimated the contribution of pPOM in comparison to formalin preserved samples treated with Folch solution (Fig. 6). The use of source data B for lipid normalized data caused an increased reliance on pPOM of the same magnitude as for not normalized data (results for source data B not shown).

**4 Discussion**

We report diverse effects of different preservation and pre-treatment methods on $\delta^{13}C$ and $\delta^{15}N$ ratios of the two studied species: *Limecola balthica* and *Crangon crangon*. The effects of preservation, lipid removal, and carbonate removal were relatively large for $\delta^{13}C$ values, while the absolute effect on $\delta^{15}N$ values was generally small. This is in agreement with previous studies that consequently highlighted that the methodological approach should be adjusted to optimize $\delta^{13}C$ (Lau et
al., 2012; Søreide et al., 2006). However, in contrast to previous studies, Bayesian mixing models were used in this study to assess impacts on the results of data analyses. Modelling results highlighted that preservation and pre-treatment effects were variable and not directly linked to the strength of the absolute change of $\delta^{13}C$ and $\delta^{15}N$ signatures. Particularly, small changes in $\delta^{15}N$ introduced by different preservation and pre-treatments (this study, Lau et al., 2012) display a so far unrecognized source of error. The observed preservation effects were strongest for formalin preserved samples, but also
freezing, which is currently thought to be less invasive method - had a clear impact on modelling results. In addition to effects on fauna's SI signatures, freezing also affected the $\delta^{15}N$ values of SOM, which introduced another alteration of mixing model result. This effect of freezing is in strong contrast to the current believe that freezing should be used as the best method to preserve samples for SIA (Bosley and Wainright, 1999; Edwards et al., 2002).





## 4.1 Diet composition of *Limecola balthica* and *Crangon crangon*

It is well documented that *L. balthica* is capable of suspension feeding as well as deposit feeding (Hummel, 1985; Törnroos et al., 2015) and accordingly a reliance on pPOM and SOM seem equally possible. It has, however, been shown that suspension feeding allows for deeper burial than deposit feeding, since the clam does not have to extend its inhalant syphon over the sediment surface (Hummel, 1985; Lin and Hines, 1994). Accordingly, suspension feeding provides better protection from predation and is the preferred feeding mode in situations with abundant food suspended in the water and epibenthic

predators, like *C. crangon*, present (Kamermans and Huitema, 1994; Lin and Hines, 1994). Consequently, a high reliance of *L. balthica*, as indicated by mixing model for dried samples, is likely, since sampling was conducted towards the end of the spring bloom and suspended OM was abundant in the sampling area.

Mixing models' results suggest a strong reliance of *C. crangon* on SOM across all preservation and pre-treatment methods' combinations. The only exception were samples treated with Folch solution but not with HCl, which showed a trend towards

a considerable reliance on pPOM (Fig. 6). Furthermore, the reliance of *C. crangon* on pPOM was around 15% higher when source data B (frozen sediment) was used as opposed to source data A (dried sediment). The overall high reliance of *C. crangon* on SOM is in agreement with previous studies that describe this species as a generalist benthic carnivore (Oh et al., 2001; Pihl and Rosenberg, 1984). For the smaller size classes of *C. crangon* (similar to the individuals in this study) it is likely that meiofauna constitutes a large proportion of their prey (Feller, 2006; Pihl and Rosenberg, 1984). Meiofaunal taxa

are often specialized on consumption of detritus, bacteria or microphytobenthos (Carman and Fry, 2002) and consequently a high indirect reliance of *C. crangon* on SOM seems likely.

Organic matter from the Vistula river (rPOM) was not identified as a considerable OM source in the diet of both species. Sampling during the spring bloom (rich in marine production) and a long period with low river inflow during winter (low input of rPOM) support such a high reliance of fauna on pPOM and SOM. Nonetheless, it is important to point out that the

low contribution of rPOM to the diet cannot be interpreted as an indication that the studied species do not consume rPOM. The reason for this interpretation is related to the fact that both pPOM and SOM that were collected at the study locations are potentially mixtures of autochthonous (marine OM) and allochthonous (rPOM) production (Maksymowska et al., 2000). The study location is influenced by the outflow of Vistula river and accordingly the OM collected at the study location cannot be considered entirely free of rPOM. Consequently, the strong contribution of SOM and pPOM to the diet of *C. crangon* and *L.*

*balthica*, respectively, demonstrates that primary consumers do not select for rPOM. However, the preference of *L. balthica* is for suspended and possibly fresh OM while *C. crangon* preys on organisms that rely on SOM.

## 4.2 Preservation and pre-treatment effects

Since dehydration of samples is an essential step in SIA for all preservation methods, sample preservation through immediate drying was considered the least invasive approach. Consequently, modelling results acquired from dried fauna



samples with source data set A (dried SOM) were considered to reflect the diet composition most accurately and strong deviations of other preservation methods from this result were interpreted as poor performance.

Mixing model results highlighted that frozen and formalin preserved *L. balthica* samples performed poorly. Formalin preservation performed worse than freezing, but the results suggest that both preservation methods should be avoided. Particularly the impact of freezing on modelling outcomes was unexpected, since samples are commonly frozen in other studies that use Bayesian mixing models (e.g. Jankowska et al., 2018). Our modelling results for frozen acidified *L. balthica* with source data B (median estimated contribution of SOM: 29%) was similar to results in a previous study in the close-by Puck lagoon that estimated 29% and 37% contribution of SOM to *L. balthica* diet during spring (Marcelina et al., 2018). Marcelina et al. (2018) preserved their samples by freezing and also acidified samples prior to SIA. Consequently, we suggest that the contribution of SOM might be overestimated in their study and the development of a mathematical correction for SIs for frozen samples, similar to the existing corrections for formalin preservation, might seem advisable. However, since freezing causes only small, taxon specific and often not significant changes in SI ratios (this study, Bosley and Wainright, 1999; Wolf et al., 2016) a possible development of any mathematical correction for frozen samples seems unlikely. The potential mechanisms behind isotopic fractionation due to freezing are not well understood. Most commonly authors that observed isotopic changes after freezing explain it by a potential damage to cells and tissues that can cause a loss of fluids, which might be isotopically enriched (Feuchtmayr and Grey, 2003; Syväranta et al., 2011; Wolf et al., 2016). However, no study has measured whether fluids released through freezing and thawing have a different isotopic composition as the remainder of the material (Wolf et al., 2016). Freezing and formalin preservation also affected the mixing model outcome for *C. crangon*. The magnitude of change, however, was rather low and the choice of preservation method appeared unimportant for this species in the present study.

Lipid contents are typically low during spring (Hill et al., 1992). Furthermore, *L. balthica* tissues were removed from their carbonate shells and the effect of carbonate removal on *C. crangon* isotopic composition has previously been shown to be not significant (Jaschinski et al., 2008). Accordingly, neither pre-treatment should have been necessary for the fauna in this study. Nonetheless, both pre-treatments applied in our study had significant effects on at least one isotope for both species and modelling outcomes were affected by both pre-treatments. The treatment where Folch solution was used for lipid removal consistently led to an increased contribution of pPOM in all models. This result can be explained by the fact that lipids are depleted in $^{13}$C (Søreide et al., 2006) and thus lipid removal caused higher $\delta^{13}$C signatures in all our samples. Since pPOM was more enriched in $^{13}$C than SOM, this might have caused a shift of the modelling results towards pPOM.

The effect of HCl treatment on $\delta^{13}$C, $\delta^{15}$N, and modelling outcomes was weaker but more variable than lipid removal. For most model runs, however, HCl treatment appears to counteract the effect of lipid removal. Somehow dried *L. balthica* and *C. crangon* (all preservation methods) isotopic ratios were affected by the pre-treatment with Folch solution followed by pre-treatment with HCl in a way that modelling results resembled the results for fauna that did not receive any pre-treatment.

Mechanisms behind isotopic fractionation due to acidification are diverse (Schlacher and Connolly, 2014) and accordingly the causes for the isotopic ratios and modelling results in this study are difficult to address. However, the species-specific





effect of acidification on $\delta^{15}N$ and the consistent effect of acidification on modelling results for *C. crangon* indicate a

potential role of chitin in the exoskeleton of crustaceans. Serrano et al. (2008) argued that HCl treatment of crustaceans may cause a shift of $\delta^{13}C$ and $\delta^{15}N$ through the known C and N loss due to hydrolytic depolymerization and deacetylation of chitin caused by the acidification (Percot et al., 2003). Although this has not been tested (Schlacher and Connolly, 2014), it provides a possible explanation for the species-specific acidification effects observed in this study.

### 4.3 Mathematically corrected data

In comparison to dried samples, $\delta^{13}C$ values were depleted after formalin preservation on average by 1.1‰ and 1.4‰ for *L. balthica* and *C. crangon*, respectively. The depletion in comparison to frozen samples was slightly lower with 0.7‰ and 1.1‰, respectively. Consequently, a general improvement of modelling results was expected after formalin preserved samples were corrected by addition of 1.28‰, since average $\delta^{13}C$ after correction was within the range of dried samples. This assumption was, however, only met for *L. balthica*, while *C. crangon* modelling results were worsened by the

correction. In contrast to the low reliance of *C. crangon* on pPOM for all preservation methods, the estimated reliance on pPOM increased after formalin correction (Fig. 6). Since $\delta^{13}C$ values were much closer to dried samples after the correction, the small insignificant (according to post-hoc test) species-specific effect of formalin preservation on $\delta^{15}N$ values is likely responsible for the contrasting effect of formalin correction on the model outcomes. $\delta^{15}N$ values of *L. balthica* were enriched on average by less than 0.1‰ and that of *C. crangon* by 0.3‰. These enrichments were comparable to the majority of

formalin induced changes to $\delta^{15}N$ reported in the literature (Lau et al., 2012). Our results for *C. crangon*, however, demonstrate that the correction of $\delta^{13}C$ only cannot compensate for a preservation effect on both SIs, even if the effect on $\delta^{15}N$ is small. For *L. balthica*, on the other hand, with the preservation effect on $\delta^{15}N$ being close to 0, the correction of $\delta^{13}C$ achieved an improvement of modelling results.

Our analysis revealed that different lipid normalization formulas for $\delta^{13}C$ should be selected for the two studied species.

Today, the common approach to lipid normalization, however, is to apply one formula across all taxa when whole food webs are studied (e.g. Silberberger et al., 2018). This approach is very likely to introduce a considerable bias in such studies, considering how differently *L. balthica* and *C. crangon* are affected. Based on our results it has to be concluded that normalization according to Post et al. (2007) should be avoided for bivalves. The data used to develop this normalization formula was largely dominated by fish, but also included multiple arthropod species. In addition, the data contained a single

sample of a unionid freshwater mussel. This single sample, however, seems to be an outlier in the data and can be perfectly corrected by the formula of McConnaughey and McRoy (1979) (Post et al., 2007).

Many bivalve species, including *L. balthica*, can store energy in the form of glycogen (Patterson and Carmichael, 2016; Wenne and Styczyńska-Jurewicz, 1987). Patterson and Carmichael (2016) showed that high glycogen content in oysters can cause C:N ratios greater than 3.5, even when lipid content is low. In contrast to lipids, glycogen is not depleted in $\delta^{13}C$

(Patterson and Carmichael, 2016). Accordingly, lipid extraction is not necessary and lipid normalization has to be avoided when glycogen is the major form of energy storage. *Limecola balthica* (among other bivalve species), however, is known to





store energy in the form of lipids as well as glycogen (Wenne and Styczyńska-Jurewicz, 1987). It is therefore impossible to know whether high C:N ratios are an indication of high lipid or glycogen content. Furthermore, glycogen content can exceed the lipid content in other benthic invertebrate taxa (e.g. the polychaete *Hediste diversicolor* (Durou et al., 2005)).

Consequently, we suggest that lipid normalization of $\delta^{13}C$ based on C:N ratios should be abandoned for bivalves and other taxa with potential energy storage other than lipids.

Even though the effect of both lipid normalization methods on $\delta^{13}C$ ratios differed from each other, mixing model results for both methods resembled each other closely for dried and frozen samples, but not for formalin preserved samples (Fig. 6). The modelling results after lipid normalization, however, were not consistently resembling the results of samples that have

been treated with Folch solution. For dried samples of both species, lipid normalization had a very limited and overall positive or neutral effect on modelling results. For frozen samples, however, lipid normalization did not improve the results but lead to completely different outcome (especially for *L. balthica*). This confirmed the results from a previous study that found lipid normalization to affect mixing model outcomes in comparison to lipid intact samples (Arostegui et al., 2019). In contrast to Arostegui et al. (2019), however, the present study was able to compare lipid normalized also with lipid extracted

samples. As a conclusion at this point, lipid normalization is not advisable for mixing models. This statement relates, however, to the fact that lipid normalization does not reproduce model results of lipid extracted samples and does not solve the question of whether lipid treatment is advisable to answer a specific ecological question.

## 5 Conclusions

Our study confirmed the results of previous studies that the absolute preservation and pre-treatment effects on $\delta^{13}C$ are high,
while effects on $\delta^{15}N$ are rather low. The strength of these effects is, however, not directly linked to their influence on the performance of Bayesian mixing models and consequently previously developed guidelines about sample preservation and pre-treatment need to be re-evaluated for this purpose. Furthermore, many observed effects on SI ratios and model performance were species-specific in this study and consequently every preservation and pre-treatment choice (for fauna and OM sources) may bias ecological interpretation. We assume that this has a potentially strong impact in the context of

multispecies studies and suggest that future research should address how this might bias studies of whole food webs.

Furthermore, our analysis has demonstrated that mathematical correction of $\delta^{13}C$ for high lipid content or formalin preservation, should be avoided or at least be applied with extreme caution if data is used in Bayesian mixing models. Even if $\delta^{13}C$ is corrected adequately, the model outcomes are only rarely improved and equally often worsened. Consequently, we recommend that the overall small, but apparently species-specific preservation and pre-treatment effects on $\delta^{15}N$ should not

be neglected in ecological studies that use stable isotope ratios to study diet composition or food-web structure.

We call for increased efforts to improve our understanding on how our methodological choices in SIA affect our interpretation of the data, since the previous focus on quantifying the absolute changes of individual SI ratios does not allow for the ideal method selection.



**Data availability.** Data will be made available to reviewers on request. Upon acceptance of the publication, data will be permanently stored within the IOPAN data system and a DOI will be assigned through DataCite.

**Author contributions.** All authors designed the research. MJS and KKM carried out the fieldwork and performed the laboratory analyses. MJS preformed the statistical analysis and drafted the manuscript. All authors interpreted the data, provided comments during writing, and proof-read the final manuscript. MK acquired funding that made this study possible.

**Competing interests.** The authors declare that they have no conflict of interest.

**Acknowledgements.** We want to thank Captain and crew of s/y Oceania. Thanks to P Makuch, M Szczepanek and Z Borawska for help during sampling. Furthermore, we want to thank B Szymczycha for guidance for chemical lipid removal and E Jordà Molina for drawings of species used in the figures.

**Financial support.** This study was funded by Polish National Science Centre, project no 2017/26/E/NZ8/00496
(COmEBACk). Sample collection was additionally supported through statutory funds of IOPAN. Katarzyna Koziorowska-Makuch's participation in this study was further supported by the Foundation for Polish Science (FNP).

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


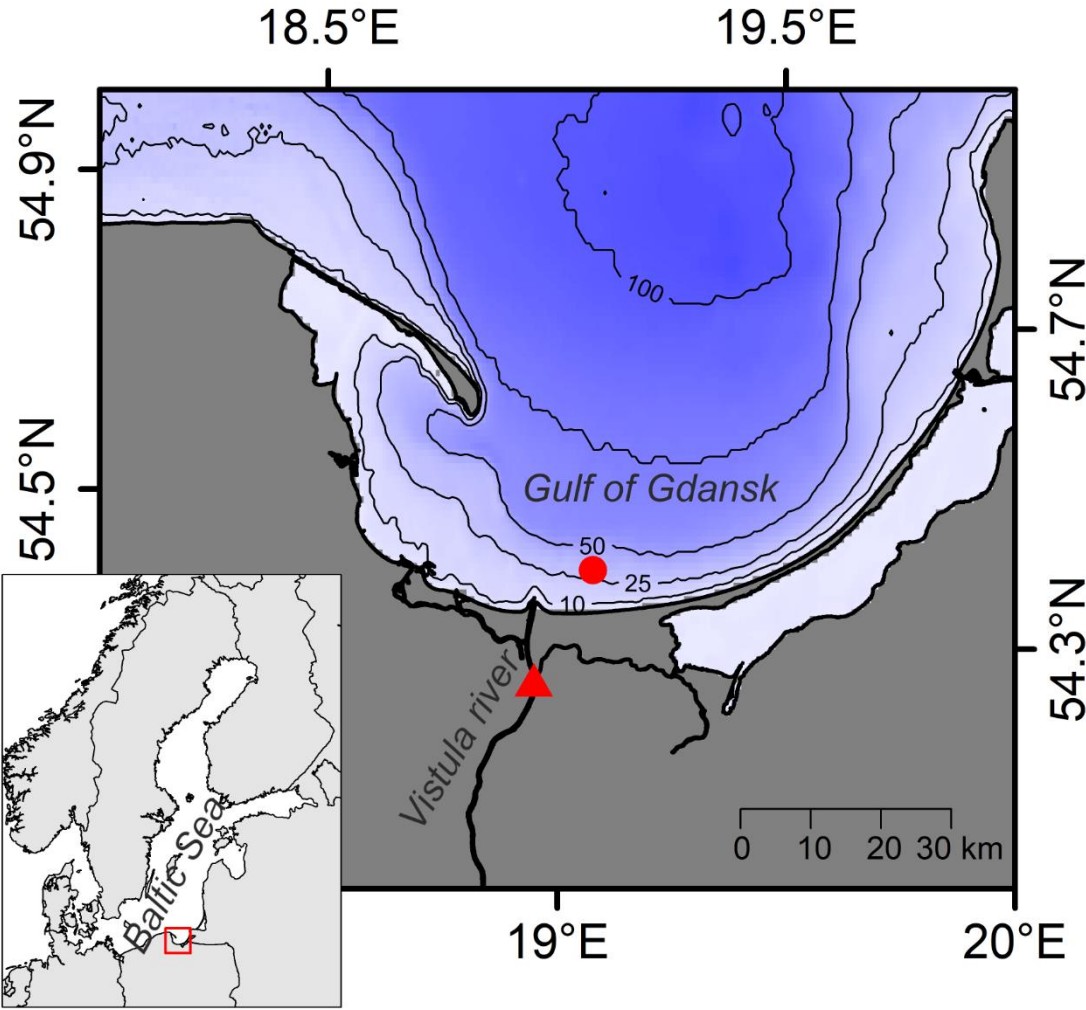

**Figure 1: Map of the study area. Sampling locations in the Gulf of Gdansk (circle) and at the Vistula river (triangle) are shown.**
**Inset: Map of the Baltic Sea region with location of the study area indicated.**





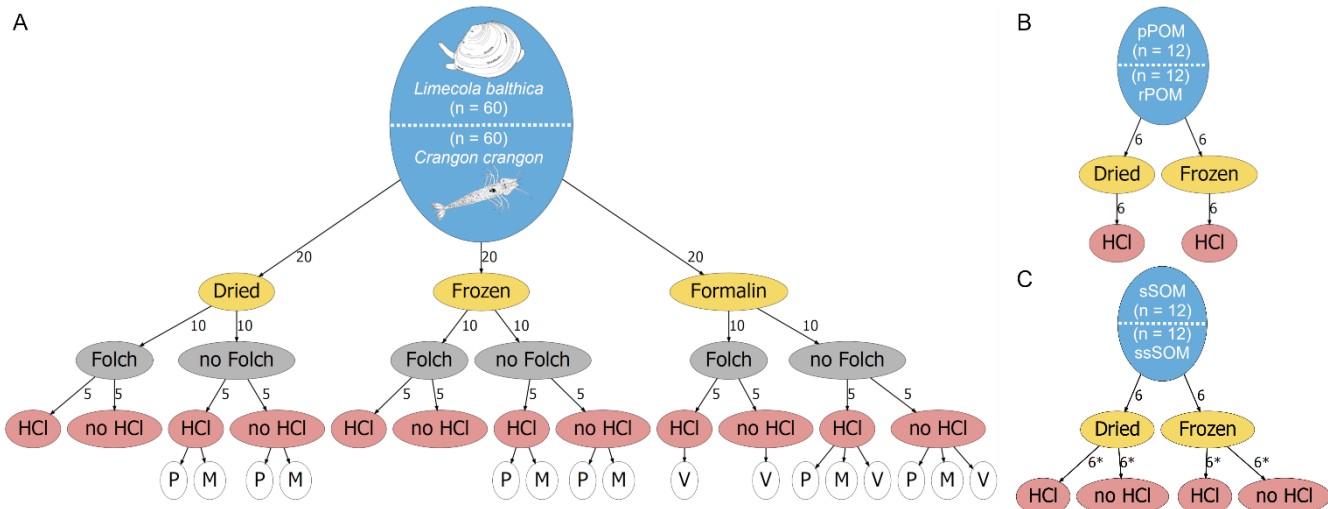

**Figure 2: Schematic overview of the study design. The total number of samples for each sample type is given in the blue ellipses for (A) Fauna, (B) SOM, and (C) POM. Preservation methods are given in yellow ellipses, lipid removal in Folch solution is shown in grey ellipses, and acidification with HCl is indicated in red ellipses. Numbers along the arrows indicate the number of samples that were used for a specific treatment. Asterix (*) indicates samples that were analysed as duplicates. White ellipses in (A) indicate which data was used for various mathematical corrections. P and M indicate lipid normalization according to Post et al. (2007) and McConnaughey and McRoy (1979), respectively. V indicates formalin correction according to Vander Zanden et al. (2003).**



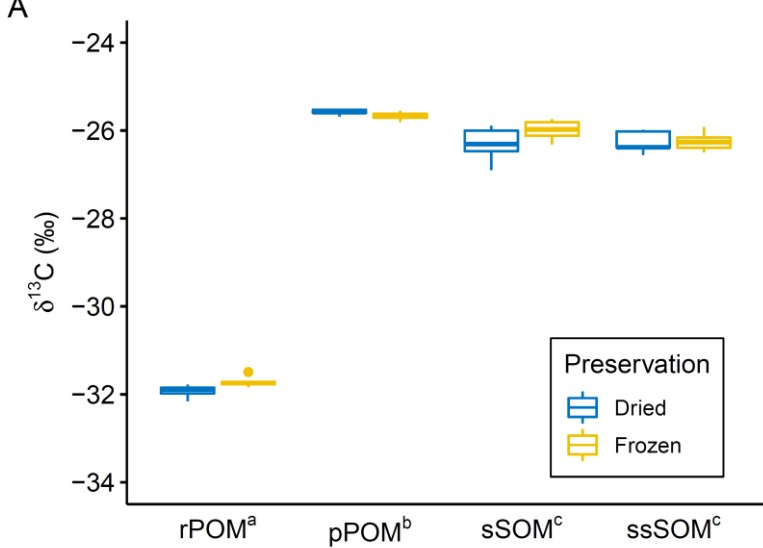

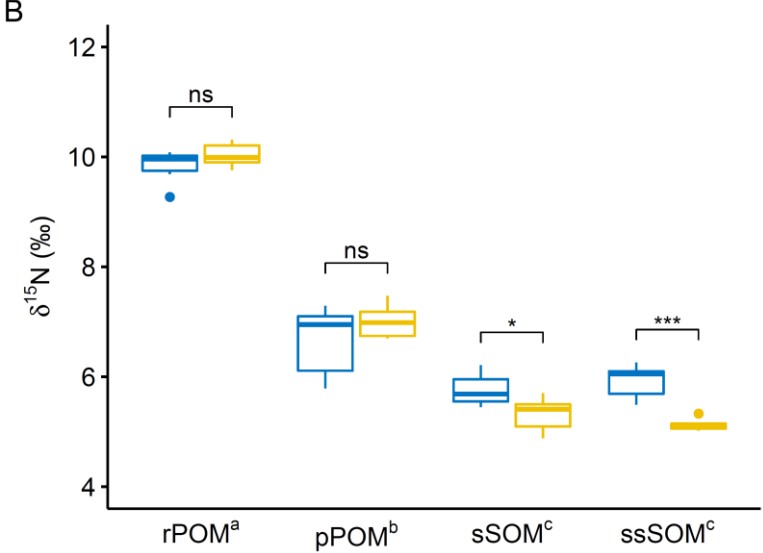


**Figure 3: Boxplot showing (A) $\delta^{13}C$ and (B) $\delta^{15}N$ distribution for differently preserved OM sources through their quartiles. Points indicate weak outliers (i.e. outliers that did not cause violation of ANOVA assumptions). Extreme outliers (rPOM$_{frozen}$: $\delta^{13}C$ = -33.27, $\delta^{15}N$ = 10.49; pPOM$_{dried}$: $\delta^{13}C$ = -26.65, $\delta^{15}N$ = 6.11; ssSOM$_{dried}$: $\delta^{13}C$ -26.44, $\delta^{15}N$ = 4.30) that were excluded from the analysis are not represented. Superscript letters indicate significantly different OM sources (i.e. OM sources with same letter are**
**not significantly different from each other). Black brackets in (B) indicate significant interaction effect between organic matter source and preservation method as identified by post-hoc test (significance levels: n.s. – not significant; * – p < 0.05; *** – p < 0.001). No brackets are shown in (A) since the interaction effect was not significant according to ANOVA and no post-hoc test was conducted.**





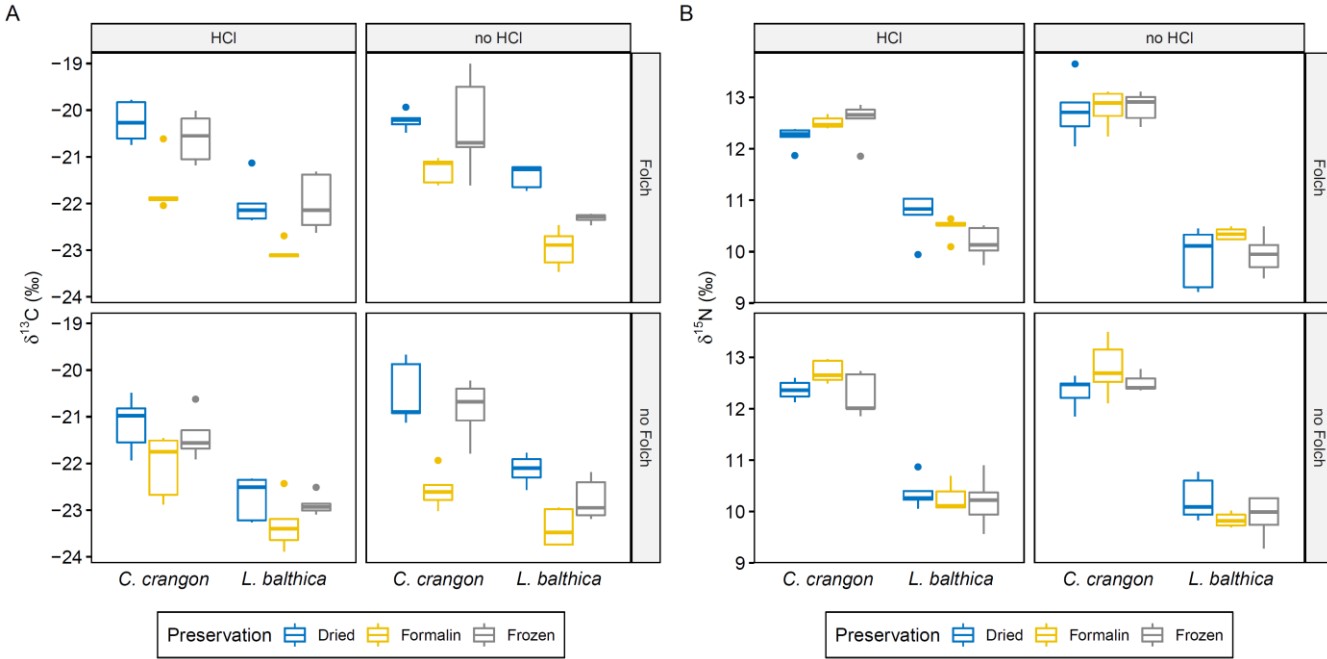

**Figure 4: Boxplots showing (A) $\delta^{13}C$ and (B) $\delta^{15}N$ distribution for all preservation pre-treatment combinations for the two collected faunal species through their quartiles. Pre-treatment abbreviations: HCl – carbonate removal with hydrochloric acid; Folch – Lipid removal with Folch solution. Significant differences between groups are given in Table 1.**






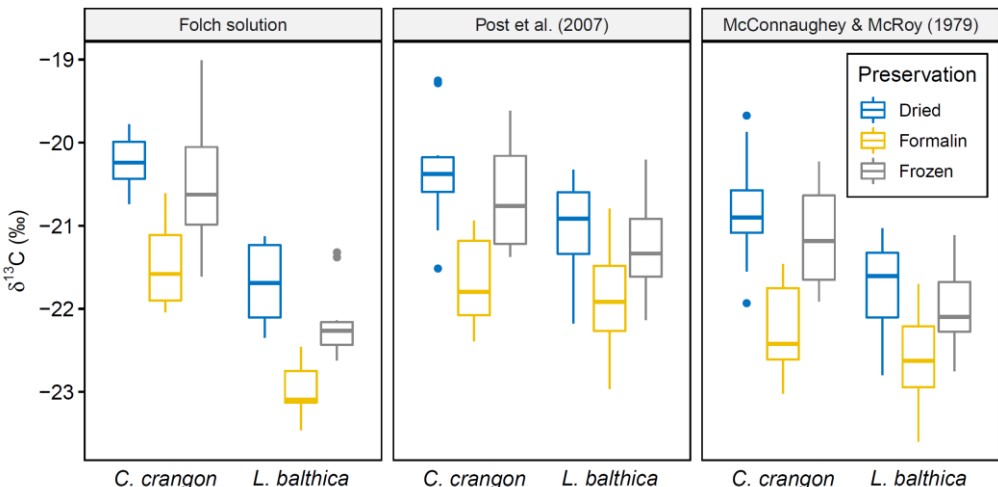

**Figure 5: Boxplot showing δ¹³C distribution of samples with lipids removed chemically (Folch solution) and mathematical lipid normalization according to Post et al. (2007) and McConnaughey & McRoy (1979) of samples not treated with Folch solution. Preservation method is indicated. Acidified and non-acidified samples were combined for this figure.**










**Figure 6: MixSIAR results - Posterior distributions for *L. balthica* and *C. crangon* diet proportions (species identity indicated for each model outcome). Source data used in the models are indicated by grey box for each row. Colours indicate the applied pre-treatments. (A) Results for dried, frozen, and formalin preserved fauna are shown together with formalin corrected data according to Vander Zanden et al. (2003). (B) Results for lipid normalized data (only results for source data A shown). Boxplots depict median (horizontal line) with 50% (box) and 95% credible intervals (vertical line).**




**Table 1: Results of factorial ANOVA for preservation and pre-treatment effects on a) $\delta^{13}$C and b) $\delta^{15}$N of fauna samples. ANOVA results for $\delta^{13}$C with lipid removed vs. lipid normalized data are given for normalization according to c) Post et al. (2007) and d) McConnaughey and McRoy (1979). Only significant effects are reported (i.e. p < 0.05).**

| Isotope | Effect | DFn | DFd | F | p-value |
|---|---|---|---|---|---|
| a) | | | | | |
| $\delta^{13}$C | Species | 1 | 96 | 271.29 | <0.001 |
| $\delta^{13}$C | Preservation | 2 | 96 | 64.86 | <0.001 |
| $\delta^{13}$C | Lipid removal | 1 | 96 | 49.40 | <0.001 |
| $\delta^{13}$C | Carbonate removal | 1 | 96 | 4.71 | 0.032 |
| b) | | | | | |
| $\delta^{15}$N | Species | 1 | 96 | 1298.62 | <0.001 |
| $\delta^{15}$N | Species:Carbonate removal | 1 | 96 | 18.466 | <0.001 |
| $\delta^{15}$N | Species:Preservation | 2 | 96 | 3.30 | 0.041 |
| c) | | | | | |
| $\delta^{13}$C | Species | 1 | 96 | 116.02 | <0.001 |
| $\delta^{13}$C | Preservation | 2 | 96 | 55.37 | <0.001 |
| $\delta^{13}$C | Lipid approach | 1 | 96 | 13.28 | <0.001 |
| $\delta^{13}$C | Carbonate removal | 1 | 96 | 13.15 | <0.001 |
| $\delta^{13}$C | Species:Lipid approach | 1 | 96 | 29.16 | <0.001 |
| d) | | | | | |
| $\delta^{13}$C | Species | 1 | 96 | 140.61 | <0.001 |
| $\delta^{13}$C | Preservation | 2 | 96 | 58.31 | <0.001 |
| $\delta^{13}$C | Lipid approach | 1 | 96 | 6.97 | 0.01 |
| $\delta^{13}$C | Carbonate removal | 1 | 96 | 9.57 | 0.003 |
| $\delta^{13}$C | Species:Lipid approach | 1 | 96 | 19.31 | <0.001 |




**Table 2: C:N ratio of the studied species with and without chemical lipid removal. Relative lipid content of individuals that had lipids chemically removed is given (i.e. weight loss during lipid removal). Given values are the mean with standard deviation.**

|  | *Limecola balthica* | *Crangon crangon* |
|---|---|---|
| C:N ratio (no lipid removal) | 4.83 (SD 0.43) | 3.89 (SD 0.15) |
| C:N ratio (lipid removal) | 4.10 (SD 0.35) | 3.39 (SD 0.13) |
| Lipid content [%] | 10.3 (SD 4.2) | 18.4 (SD 3.9) |