# Peer review of "Sample preservation and pre-treatment in stable isotope analysis: Implications for the study of aquatic food webs"

_Biogeosciences, 2020_

## Referee Comment (RC1) · Anonymous Referee #1 · 8 Jun 2020

The ms by Silberberger et al. investigated the effect of different preservation methods (drying, freezing, formalin) on C and N stable isotope ratios in two marine invertebrate species, and the effect of acidification and lipid removal. The authors then apply Bayesian mixing models to determine the extent to which these sample prep methods affect the outcome of the mixing models.

While there are some valuable data in the manuscript, I fail to see the overall relevance of explicitly investigating how these sample prep differences translate into mixing model output. The point is that this will depend strongly on the 'absolute' values of the sources and consumers, and I don't see how general conclusions can be drawn. The

authors mention that they also tested whether lipid and formalin corrections 'improved' the model results but there is no objective way to claim whether the output is better or worse then the original. "improved" implies closer to reality. Hence I do not see how we should interpret the authors' conclusions that (L408): 'the model outcomes are only rarely improved and equally often worsened' – the authors remain vague on how they interpret this.

If dried samples (and acidification for d13C if carbonates could be present) are the reference, then indeed all of the pre-treatment and preservation methods may or may not results in shifts in measured d13C or d15N values. A wealth of studies have assessed such changes, often with variable outcomes – but we have a good idea of the range of changes that can be expected. When subsequently applying a (Bayesian or not) mixing model, the key assumptions that need to be be made are regarding the trophic shifts for d13C and d15N. In my view, it would make sense to think about how these trophic shifts were determined – if those literature estimates were based on measurements with standard sample prep (no lipid extraction etc) then they need to be applied to similar data, if not then a correction to your sample data is needed in order to account for that.

If I consider the objectives of this manuscript: (i) quantify how sample preservation and pre-treatment affect carbon and nitrogen SI ratios (ii) identify potential interaction effects between preservation and pre-treatment methods (iii) study how preservation and pre-treatment affect the results of Bayesian mixing models (iv) assess whether lipid normalization and mathematical formalin correction should be used to adjust data for the use in such models.

then for me (i) and (ii) are fine, but (iii) and (iv) are not. I feel the ms should either forus on objectives (i) and (ii) but in that case it becomes a small dataset that is perhaps not sufficiently novel compared to the existing literature, or alternative think on expanding the scope – but I'm not sure that is feasible with the data at hand.

Specific suggestions/comments:

The authors apply a 'scaled trophic fractionation' for d13C, whereby this is assumed to be large for the first trophic transfer (4 +/- 1.3 per mil) and smaller (0.4 +/- 1.3 per mil) for subsequent trophic transfers. Frankly, it is the first time I come across this, and if I follow the references this is based on, this all comes from a single paper (Hobson et al. 1995). I do not see much confirmation of the validity of this assumption in later reviews on trophic fractionation, e.g. Caut et al. (2009) Caut et al (2009) Variation in discrimination factors ($\Delta$15N and $\Delta$13C): the effect of diet isotopic values and applications for diet reconstruction. Journal of Applied Ecology 46: 443-453. https://doi.org/10.1111/j.1365-2664.2009.01620.x

In particular for datasets such as the one the authors have, where C and N isotope ratios for most sources (pPOM, sSOM, ssSOM) are very similar and one is quite different (rPOM), the choice of the frationation factor has a very strong influence on the results, likely much more then the relatively small shifts induced by sample pre-treatment or storage.

There is also a large body of additional literature looking into the effects of lipid extraction and lipid corrections, e.g. the recent one by Cloyed et al. (2020) and references therein. Cloyed et al. (2020) The effects of lipid extraction on $\delta$13C and $\delta$15N values and use of lipid‐correction models across tissues, taxa and trophic groups. Methods in Ecology & Evolution 11: 751-762. https://doi.org/10.1111/2041-210X.13386

Terminology:

-avoid the use of 'stable isotope concentrations' (L38), you're looking at stable isotope ratios

-d13C values of different samples are higher or lower compared to each other, or they increase or decrease. Samples are enriched or depleted in 13C relative to each other. Avoid the use of mixed terminology such as 'a depletion in d13C' (L43 and throughout

the ms), an 'enrichment in d13C' (L43 and throughout the ms).

-data corrections: in the methods (L131-133) the authors mention that data were corrected using pure reference gases calibrated against IAEA standards. This is not correct: the reference gases should not be used to calibrate the data, they merely serve as a monitoring gas, and data should be corrected using results of certified (or in-house calibrated) standards during the run. Perhaps this was done (I hope so) but it is merely not formulated properly.

-C/N ratios (L163 and further): to avoid confusion, please mention if these are weight/weight or molar ratios.

---

## Referee Comment (RC2) · Anonymous Referee #2 · 19 Jun 2020

This study reports the effects of sample preservation and further treatment on stable isotope values in two marine species (Crustacea and Bivalva) and further consequences on the application of Bayesian mixing models to infer the relative contribution of several food sources. The text "Implications for the study of aquatic food webs" in the title seems to me too ambitious considering that the study only refers to two species. The first part of the study in well-presented and organized, providing interesting information on SI for the focussed species. The authors use different transformations and corrections on original Si values, depending on previous treatment of the samples. So many data manipulations seem rather confusing sometimes and will certainly affect the results achieved in the second part of the manuscript, focussed on mixing models. I

really do not understand the interest of comparing the author's data on SI values with those resulting from mathematical corrections according to Post et al. (2007) and Mc-Connaughey & McRoy (1979). From my point of view and considering the huge work made on samples treatments, the original data provided in the study is of certain value in itself and will partially contribute to increase the knowledge on stable isotopes in marine organisms. The most interesting part of the study would have been that the authors provide mathematical corrections for the conversion between preservation methods for the objective species, which should be species-specific. About the second part of the study, there are several constrains. The first one refers to the values applied to trophic enrichment factors (TEF) for d13C and d15N. Generalist TEFs are useful when we do not know the actual TEF for a given species, which is the case of this study. However, in some cases those TEFs are unrealistic for a given species, e.g. Planas et al. (2020). (Effect of diet on breeders and inheritance in syngnathids: application of isotopic experimentally derived data to field studies. MEPS https://doi.org/10.3354/meps13315). Hence, the use of unspecific TEF values should not be used as reference for checking "improvement" of mixing model outputs. The use of the term "improvement" is another issue in this study. As the authors claim, an objective of the mixing models analysis performed was to assess whether normalization and mathematical corrections should be used to adjust data for the use in such models and to improve modelling results. Achieving an improved model imply a comparison with a control model, which is lacking in the study. The use of generalist TFEs do not ensure that model outputs are actual references for comparisons. Besides, mixing models provide relative contribution estimates of dietary sources. Ideally, overlapping significance of estimates should be analyzed (Bhattacharyya's coefficients) to assess the accuracy of source contribution estimates (Bhattacharyya, A., 1943. On a measure of divergence between two statistical populations defined by their probability distributions. Bull Calcutta Math Soc 35:99-109). Mixing models were carried out using a "long" run approach (which is not always sufficiently long as expected). Even though diagnostics (Gelman-Rubin and Geweke) might be fine with a long run, important differences in outputs might arise

sometimes when the model is submitted to an "extreme" run. Did the authors assayed any extreme run? If so, were there any differences compared to a long run? Finally, the use of isotopic "depletion" or "enrichment" is not correct when referring to values obtained from sample treatments. The terms "decrease" or "increase" must be used instead.

---

## Author Comment (AC1) · 9 Jul 2020

We would like to thank you for your insightful comments regarding our manuscript. Based on them and the review of Referee #2, we have decided to make the following major adaptations to our manuscript:

- We will add a paragraph in the introduction about recent Bayesian Stable Isotope mixing models (specifically the MixSIAR framework) to justify our objective of addressing the impact of preservation and pre-treatment effects on such models. Here we will explain that these models utilize mean and variance information for source, consumer, and trophic enrichment factors, point out that Bayesian mixing models are multivariate analyses and that newer models also incorporate tracer covariance, since carbon and nitrogen isotopic ratios are often coupled (Hopkins & Ferguson, 2012, Parnell et al. 2013, Stock et al. 2018). However, studies that address preservation and pre-treatment effects on stable isotope ratios treat each isotope independently and accordingly there is a potential for biases in mixing models that is not recognized yet.

- Furthermore, we will clarify our interest in the comparison with lipid normalization and formalin correction from literature data. Our main motivation lies in the fact that such general corrections are widely applied to invertebrates, because species-specific or even tissue-specific corrections are typically only available for vertebrates. A recent review of Arostegui et al. (2019) showed that many mixing model studies use lipid corrected data. They demonstrated that estimated diet proportions can be strongly affected by lipid correction in comparison to lipid intact samples. However, since lipid corrections are typically applied when no comparison to lipid extracted samples is possible, we consider it important to identify whether the adjustment translates into a similar model outcome as lipid extracted samples. And accordingly for formalin correction.

- We have re-evaluated the trophic enrichment factors we used in our models. We came to the conclusion that the chosen trophic fractionation for $\delta$15N (3.4$\pm$1) should be kept. Also we think the trophic fractionation for the first trophic step from POM/SOM to primary consumer for $\delta$13C (4.0$\pm$1.3) is a good choice. For the second trophic step, however, we have decided to adjust the TEF for $\delta$13C to 0.8$\pm$0.5 according to Antonio et al. (2011) who reported differences between $\delta$13C ratios for Crangon uritai and those of a variety of prey species (known from gut contents) in the range of 0.3-1.3‰ So far, we ran mixing models with this new TEF for an initial assessment on short setting and it appears that this change will not affect our results strongly. However, if our results will be affected by this after full model runs, the manuscript will be adapted. Furthermore, we will add a more detailed justification for all the chosen TEF in the method section.

- We will also add a section about TEF in the discussion, where we will compare our observed fractionation to the chosen TEFs and discuss the importance of them in

mixing models.

- We are currently running all models on extreme setting (this is ongoing, but due to the high number of models this will take some time). So far, we have not seen any considerable differences between the long and the extreme runs, but we will adjust our manuscript accordingly if necessary.

- We will use the Bhattacharyya Coefficient to include pairwise comparisons of the probability distributions of the source contributions.

Please find below our specific answers to your comments:

Comment: The ms by Silberberger et al. investigated the effect of different preservation methods (drying, freezing, formalin) on C and N stable isotope ratios in two marine invertebrate species, and the effect of acidification and lipid removal. The authors then apply Bayesian mixing models to determine the extent to which these sample prep methods affect the outcome of the mixing models. While there are some valuable data in the manuscript, I fail to see the overall relevance of explicitly investigating how these sample prep differences translate into mixing model output. The point is that this will depend strongly on the 'absolute' values of the sources and consumers, and I don't see how general conclusions can be drawn.

Response: - We have realized that our introduction did not introduce Bayesian SIMMs and accordingly the justification of objectives 3 and 4 is not given. Of course, absolute values of sources and consumers together with the chosen trophic fractionation determine the mixing model results. However, the fact that such models are multivariate models but studies that address preservation or pretreatment effects consider isotopes individually, makes it difficult to predict how any effect translates to mixing models. Furthermore, models like IsotopeR or MixSIAR framework incorporate variance of sources and mixtures but also that tracer covary in source/mixture (Hopkins & Ferguson, 2012, Parnell et al. 2013, Stock et al. 2018). This is in contrast to our understanding of preservation and pre-treatment effects, which is treating different isotopes

independently from each other and mostly worry about average absolute changes of the isotope ratios. Consequently, we think it is important to compare how sampling and sample processing affect stable isotope ratios and how this translates to mixing models. We will clarify this in the introduction prior to the objectives to justify the motivation for the mixing models.

Comment: The authors mention that they also tested whether lipid and formalin corrections 'improved' the model results but there is no objective way to claim whether the output is better or worse then the original. "improved" implies closer to reality. Hence I do not see how we should interpret the authors' conclusions that (L408): 'the model outcomes are only rarely improved and equally often worsened' – the authors remain vague on how they interpret this.

Response: - With regard to lipid and formalin correction, the word 'improved' was used to describe that the modeling result was closer to the modeling results of the treatment for which the correction was applied. The purpose of these corrections is quite clear, e.g. lipid normalization is applied to account for not removing lipids from samples and the desired result after lipid normalization would resemble the result for samples that had lipid removed. We will adjust the ms to clarify what is the desired outcome after corrections and we will avoid using the term "improved". Also, we will use Bhattacharyya Coefficient to make pairwise comparisons of probability distributions, which will help to clarify the results section and discussion as it provides an objective way of identifying significant overlap in diet contribution between models (in addition to the visual representation of the credible intervals).

Comment: If dried samples (and acidification for d13C if carbonates could be present) are the reference, then indeed all of the pre-treatment and preservation methods may or may not results in shifts in measured d13C or d15N values. A wealth of studies have assessed such changes, often with variable outcomes – but we have a good idea of the range of changes that can be expected. When subsequently applying a (Bayesian or not) mixing model, the key assumptions that need to be made are regarding the trophic

shifts for d13C and d15N.

Response: - We agree that an important assumption for mixing models are the TEFs. This issue is rather well recognized (e.g. Bond & Diamond 2011). Any mixing model basically requires the user to make an educated guess based on the literature or experimentally determine new TEF. Recent Bayesian mixing models (in contrast to earlier mixing models), however, include uncertainty in TEFs into Bayesian mixing models (Stock and Semmens 2016) and accordingly are somehow accounting for this general difficulty in mixing models, if appropriate model parameters are selected. Stock and Semmens (2016) have pointed out that they "suspect mixing model users tend to underestimate TDF variance when using borrowed values". We believe we decided for reasonable TEFs and use appropriate variance for them (please see detailed justification for the chosen TEF below). Nonetheless, we acknowledge that there remains always some uncertainty about TEFs and will give this aspect some space in the manuscript. However, we argue that the choice of the TEFs is not as essential for the comparison of different models in our study since our focus is on the effects of preservation and pre-treatment. We used the same TEFs across all models and accordingly all the models are equally right or wrong and the differences among the models are caused by the preservation and treatment. Since trophic enrichment is a natural process and has happened prior to sample collection, it is appropriate to use the same trophic fractionation across all models. Differences between them were introduced by our methodological choices.

Comment: In my view, it would make sense to think about how these trophic shifts were determined – if those literature estimates were based on measurements with standard sample prep (no lipid extraction etc) then they need to be applied to similar data, if not then a correction to your sample data is needed in order to account for that.

Response: - We agree that it would make sense to think about this, but: o this requires prior knowledge about how preservation and pre-treatment effects on individual isotopes translate into the results of MixSIAR. (objective 3) o this also requires prior

knowledge about whether the applied corrections will give us the same modelling results as for the treatment for which it corrects for (objective 4) If we were able to get model results after a data correction that resembles the model result for samples that were treated accordingly, only then it would make sense to think about such an approach. - We will discuss the aspect of how trophic fractionation is determined in the literature, and specifically for the TEFs we used. However, since our corrections did not consistently achieve the desired outcome, we will advise against this approach (At least for benthic invertebrates for which species-specific correction methods are virtually not existing)

Comment: If I consider the objectives of this manuscript: (i) quantify how sample preservation and pre-treatment affect carbon and nitrogen SI ratios (ii) identify potential interaction effects between preservation and pre-treatment methods (iii) study how preservation and pre-treatment affect the results of Bayesian mixing models (iv) assess whether lipid normalization and mathematical formalin correction should be used to adjust data for the use in such models. then for me (i) and (ii) are fine, but (iii) and (iv) are not. I feel the ms should either forus on objectives (i) and (ii) but in that case it becomes a small dataset that is perhaps not sufficiently novel compared to the existing literature, or alternative think on expanding the scope – but I'm not sure that is feasible with the data at hand.

Response: - As mentioned above, we have realized that our introduction failed to give a clear justification for objectives 3 and 4 and we will edit the manuscript as mentioned above. In addition, our results highlight the importance of addressing these objectives. For example, in the first part of our study, we did not detect a significant difference between dried and frozen Limecola balthica for both isotopes. Nonetheless, the mixing models differed very strongly between dried (predominantly pPOM diet) and frozen (mixed diet of pPOM and SOM). Consequently, we would not apply any correction based on the effects on the isotope ratios, but it alters the estimation of the diet considerably. We will edit the manuscript to highlight these discrepancies between the

two parts (individual isotope ratios vs. mixing models) as it is from our point of view a crucial aspect of this paper.

Specific suggestions/comments: Comment: The authors apply a 'scaled trophic fractionation' for d13C, whereby this is assumed to be large for the first trophic transfer (4 +/- 1.3 per mil) and smaller (0.4 +/- 1.3 per mil) for subsequent trophic transfers. Frankly, it is the first time I come across this, and if I follow the references this is based on, this all comes from a single paper (Hobson et al. 1995). I do not see much confirmation of the validity of this assumption in later reviews on trophic fractionation, e.g. Caut et al. (2009) Caut et al (2009) Variation in discrimination factors ($\Delta$15N and $\Delta$13C): the effect of diet isotopic values and applications for diet reconstruction. Journal of Applied Ecology 46: 443-453. https://doi.org/10.1111/j.1365-2664.2009.01620.x

Response: - Unfortunately, the trophic transfer from POM/SOM to benthic primary consumers is typically excluded from reviews (like Caut et. al (2009) or McCutchan et al. (2003)) because the diet is a mixture. We provided Hobson et al. (1995) as a reference for the trophic fractionation of 4±1.3, since we are not aware of a review that could be used for this strong trophic enrichment in 13C. Our choice is, however, not based only on this study but on multiple studies that observed this trophic enrichment for POM/SOM and benthic consumers globally. To our knowledge Fry and Sherr (1984) were the first to recognize such a global pattern for POC to benthic filter feeder's trophic enrichment ~4‰ for 13C and since then this is omnipresent in marine benthic food web studies. For example, Nerot et al. (2012) reports similar trophic enrichment of benthic filter feeders in comparison to POM along a depth gradient in the northern Bay of Biscay. Iken et al. (2010) found a similar strong enrichment from POM and SOM to benthic consumers in the Pacific Arctic under 4 different water masses with differing 13C baseline. Furthermore, a similar trophic enrichment can be deducted for the Gulf of Gdansk from the data presented in Sokołowski et al. (2012). Also, we observed the same in our own studies from the North Sea and northern Norway (Silberberger et al. 2018) or Svalbard (Kedra et al. 2012). While the reasons for

this strong fractionation seem to remain unclear, it is a general pattern throughout the literature and applying a smaller general TEF according to one of the highly cited reviews would be not appropriate here, especially with regard to our data structure that clearly shows a strong 13C enrichment from the OM sources to Limecola balthica (compare fig. 2 and 3 in our ms). We have, however, re-evaluated all our chosen fractionations and have come to the conclusion to make a slight adjustment to the TEF for $\delta$13C for Crangon crangon according to Antonio et al. (2011) who studied Crangon uritai and its prey (0.8±0.5). A widely applied trophic fractionation of 3.4±1 for nitrogen was chosen according to Post (2002). He developed this general value largely on filter-feeding bivalves (pelagic baseline) and grazing snails (littoral baseline) and accordingly we assume it a suitable trophic fractionation for Limecola balthica. The same trophic fractionation was applied for the second trophic step, since we could not find a more suitable fractionation. The applied fractionation compares well with the fractionation of 3.6‰ that has been assumed by Fry (1988) in a study that included also Crangon sp. Furthermore, it is quite close to experimentally determined trophic enrichment of a Mysid that was fed an Artemia diet (3.55) (Gorokhova & Hanssen 1999). We also considered calculating TEF from formulas for invertebrates from Caut et al. (2009). However, we decided against it for 3 reasons: (1) the calculated fractionations would be unrealistically low with regard to our data, (2) the formula only gives a mean trophic fractionation without any variance term, and (3) Caut et al. (2009) reports R2 of 0.09 for the invertebrate formula for carbon and accordingly we don't think this specific formula should be applied.

Comment: In particular for datasets such as the one the authors have, where C and N isotope ratios for most sources (pPOM, sSOM, ssSOM) are very similar and one is quite different (rPOM), the choice of the frationation factor has a very strong influence on the results, likely much more then the relatively small shifts induced by sample pre-treatment or storage.

Response: - We agree that the choice of fractionation is crucial for Bayesian stable

isotope mixing models. However, this is a well-recognized limitation (e.g. Parnell et al. 2013, Bond and Diamond 2011) and requires an educated choice based on the literature. As described above, we have given this long consideration and believe we have chosen adequate TEF. As mentioned before, since we apply the same trophic fractionation for all samples from one species, all our models are equally right or wrong and accordingly our observed differences are caused by preservation and pre-treatment. We will discuss the trophic fractionation we selected in comparison to our data and also the importance of the applied trophic fractionation in models.

Comment: There is also a large body of additional literature looking into the effects of lipid extraction and lipid corrections, e.g. the recent one by Cloyed et al. (2020) and references therein. Cloyed et al. (2020) The effects of lipid extraction on $\delta$13C and $\delta$15N values and use of lipid Ÿ Rcorrection models across tissues, taxa and trophic groups. Methods in Ecology & Evolution 11: 751-762. https://doi.org/10.1111/2041-210X.13386

Response: - Thank you for pointing out the literature. We have looked into the paper and the references in it and have found some interesting literature we will address. However, that paper also highlights that there are big differences between the knowledge available for vertebrates and invertebrates, which we will address as well.

Terminology: Comment: -avoid the use of 'stable isotope concentrations' (L38), you're looking at stable isotope ratios

Response: - We will edit the ms accordingly

Comment: -d13C values of different samples are higher or lower compared to each other, or they increase or decrease. Samples are enriched or depleted in 13C relative to each other. Avoid the use of mixed terminology such as 'a depletion in d13C' (L43 and throughout the ms), an 'enrichment in d13C' (L43 and throughout the ms).

Response: - We will edit the ms accordingly

Comment: -data corrections: in the methods (L131-133) the authors mention that data were corrected using pure reference gases calibrated against IAEA standards. This is not correct: the reference gases should not be used to calibrate the data, they merely serve as a monitoring gas, and data should be corrected using results of certified (or in-house calibrated) standards during the run. Perhaps this was done (I hope so) but it is merely not formulated properly.

Response: - Thank you for pointing this out. This was done. We will clarify the text in the revised ms.

Comment: -C/N ratios (L163 and further): to avoid confusion, please mention if these are weight/weight or molar ratios.

Response: - We will edit the ms accordingly

References:

Antonio ES, Akihide Kasai, Masahiro Ueno, Yuka Ishihi, Hisashi Yokoyama, Yoh Yamashita, Diet Shift in the Sand Shrimp Crangon Uritai along the Estuary-Marine Gradient, Journal of Crustacean Biology, Volume 31, Issue 4, 1 October 2011, Pages 635–646, https://doi.org/10.1651/10-3424.1

Arostegui, M.C., Schindler, D.E. & Holtgrieve, G.W. Does lipid-correction introduce biases into isotopic mixing models? Implications for diet reconstruction studies. Oecologia 191, 745–755 (2019). https://doi.org/10.1007/s00442-019-04525-7

Bond, A.L. and Diamond, A.W. (2011), Recent Bayesian stable‐isotope mixing models are highly sensitive to variation in discrimination factors. Ecological Applications, 21: 1017-1023. doi:10.1890/09-2409.1

Caut, S., Angulo, E. and Courchamp, F. (2009), Variation in discrimination factors ($\Delta$15N and $\Delta$13C): the effect of diet isotopic values and applications for diet reconstruction. Journal of Applied Ecology, 46: 443-453. doi:10.1111/j.1365-2664.2009.01620.x

Fry, Brian, (1988), Food web structure on Georges Bank from stable C, N, and S isotopic compositions, Limnology and Oceanography, 33, doi: 10.4319/lo.1988.33.5.1182

Fry B., Sherr E.B. (1984) $\delta$13C Measurements as Indicators of Carbon Flow in Marine and Freshwater Ecosystems. Contrib. Mar. Sci. 27: 13-47.

Gorokhova E, Sture Hansson (1999) An experimental study on variations in stable carbon and nitrogen isotope fractionation during growth of Mysis mixta and Neomysis integer, Canadian Journal of Fisheries and Aquatic Sciences 56:2203-2210, https://doi.org/10.1139/f99-149

Hobson, K. A., Ambrose, W. G. and Renaud, P. E.: Sources of primary production, benthic-pelagic coupling, and trophic relationships within the Northeast Water Polynya: insights from ?13C and ?15N analysis, Mar. Ecol. Prog. Ser., 128(1–3), 1-10, doi:10.3354/meps128001, 1995.

Hopkins JB III, Ferguson JM (2012) Estimating the Diets of Animals Using Stable Isotopes and a Comprehensive Bayesian Mixing Model. PLoS ONE 7(1): e28478. https://doi.org/10.1371/journal.pone.0028478

Iken K, Bodil Bluhm, Kenneth Dunton (2010) Benthic food-web structure under differing water mass properties in the southern Chukchi Sea, Deep Sea Research Part II: Topical Studies in Oceanography 57: 71-85, https://doi.org/10.1016/j.dsr2.2009.08.007

KÄŹdra M, Karol Kuliński, Wojciech Walkusz, Joanna LegeÅijyńska (2012) The shallow benthic food web structure in the high Arctic does not follow seasonal changes in the surrounding environment, Estuarine, Coastal and Shelf Science 114: 183-191, https://doi.org/10.1016/j.ecss.2012.08.015

McCutchan, J.H., Jr, Lewis, W.M., Jr, Kendall, C. and McGrath, C.C. (2003), Variation in trophic shift for stable isotope ratios of carbon, nitrogen, and sulfur. Oikos, 102: 378-390. doi:10.1034/j.1600-0706.2003.12098.x

Nerot C, Anne Lorrain, Jacques Grall, David P. Gillikin, Jean-Marie Munaron, Hervé Le

Bris, Yves-Marie Paulet (2012) Stable isotope variations in benthic filter feeders across a large depth gradient on the continental shelf, Estuarine, Coastal and Shelf Science 96: 228-235, https://doi.org/10.1016/j.ecss.2011.11.004

Parnell, A.C., Phillips, D.L., Bearhop, S., Semmens, B.X., Ward, E.J., Moore, J.W., Jackson, A.L., Grey, J., Kelly, D.J. and Inger, R. (2013), Bayesian stable isotope mixing models. Environmetrics, 24: 387-399. doi:10.1002/env.2221

Post, D. M.: Using stable isotopes to estimate trophic position: models, methos, and assumptions, Ecology, 83(3), 703–718, doi:Doi 10.2307/3071875, 2002

Silberberger, M. J., Renaud, P. E., Kröncke, I. and Reiss, H.: Food-web structure in four locations along the European shelf indicates spatial differences in ecosystem functioning, Front. Mar. Sci., 5(APR), doi:10.3389/fmars.2018.00119, 2018

Sokołowski A, M. Wołowicz, H. Asmus, R. Asmus, A. Carlier, Z. Gasiunaité, A. Grémare, H. Hummel, J. Lesutiené, A. Razinkovas, P.E. Renaud, P. Richard, M. KÄŹdra (2012) Is benthic food web structure related to diversity of marine macrobenthic communities?, Estuarine, Coastal and Shelf Science, 108:76-86, https://doi.org/10.1016/j.ecss.2011.11.011

Stock BC, Jackson AL, Ward EJ, Parnell AC, Phillips DL, Semmens BX. 2018. Analyzing mixing systems using a new generation of Bayesian tracer mixing models. PeerJ 6:e5096 https://doi.org/10.7717/peerj.5096 Stock, B.C. and Semmens, B.X. (2016), Unifying error structures in commonly used biotracer mixing models. Ecology, 97: 2562-2569. doi:10.1002/ecy.1517
* * *

---

## Author Comment (AC2) · 9 Jul 2020

We would like to thank you for your insightful comments regarding our manuscript. Based on them and the review of Referee #1, we have decided to make the following major adaptations to our manuscript:

- We will add a paragraph in the introduction about recent Bayesian Stable Isotope mixing models (specifically the MixSIAR framework) to justify our objective of addressing the impact of preservation and pre-treatment effects on such models. Here we will explain that these models utilize mean and variance information for source, consumer, and trophic enrichment factors, point out that Bayesian mixing models are multivariate analyses and that newer models also incorporate tracer covariance, since carbon and nitrogen isotopic ratios are often coupled (Hopkins & Ferguson, 2012, Parnell et al. 2013, Stock et al. 2018). However, studies that address preservation and pre-treatment effects on stable isotope ratios treat each isotope independently and accordingly there is a potential for biases in mixing models that is not recognized yet.

- Furthermore, we will clarify our interest in the comparison with lipid normalization and formalin correction from literature data. Our main motivation lies in the fact that such general corrections are widely applied to invertebrates, because species-specific or even tissue-specific corrections are typically only available for vertebrates. A recent review of Arostegui et al. (2019) showed that many mixing model studies use lipid corrected data. They demonstrated that estimated diet proportions can be strongly affected by lipid correction in comparison to lipid intact samples. However, since lipid corrections are typically applied when no comparison to lipid extracted samples is possible, we consider it important to identify whether the adjustment translates into a similar model outcome as lipid extracted samples. And accordingly for formalin correction.

- We have re-evaluated the trophic enrichment factors we used in our models. We came to the conclusion that the chosen trophic fractionation for $\delta$15N (3.4$\pm$1) should be kept. Also we think the trophic fractionation for the first trophic step from POM/SOM to primary consumer for $\delta$13C (4.0$\pm$1.3) is a good choice. For the second trophic step, however, we have decided to adjust the TEF for $\delta$ 13C to 0.8$\pm$0.5 according to Antonio et al. (2011) who reported differences between $\delta$13C ratios for Crangon uritai and those of a variety of prey species (known from gut contents) in the range of 0.3-1.3‰. So far, we ran mixing models with this new TEF for an initial assessment on short setting and it appears that this change will not affect our results strongly. However, if our results will be affected by this after full model runs, the manuscript will be adapted. Furthermore, we will add a more detailed justification for all the chosen TEF in the method section.

- We will also add a section about TEF in the discussion, where we will compare our observed fractionation to the chosen TEFs and discuss the importance of them in

mixing models.

- We are currently running all models on extreme setting (this is ongoing, but due to the high number of models this will take some time). So far, we have not seen any considerable differences between the long and the extreme runs, but we will adjust our manuscript accordingly if necessary.

- We will use the Bhattacharyya Coefficient to include pairwise comparisons of the probability distributions of the source contributions.

Please find below our specific answers to your comments:

Comment: This study reports the effects of sample preservation and further treatment on stable isotope values in two marine species (Crustacea and Bivalva) and further consequences on the application of Bayesian mixing models to infer the relative contribution of several food sources. The text "Implications for the study of aquatic food webs" in the title seems to me too ambitious considering that the study only refers to two species.

Response: - We are reconsidering the title. We have not made a final decision about the new title yet, but we are planning to replace the second part of the current title with a mentioning of stable isotope mixing models.

Comment: The first part of the study in well-presented and organized, providing interesting information on SI for the focused species. The authors use different transformations and corrections on original Si values, depending on previous treatment of the samples. So many data manipulations seem rather confusing sometimes and will certainly affect the results achieved in the second part of the manuscript, focused on mixing models.

Response: - We are aware of the complexity of the treatments and corrections in our study and will keep this in mind during the manuscript editing to avoid confusion for the reader as much as possible.

Comment: I really do not understand the interest of comparing the author's data on SI values with those resulting from mathematical corrections according to Post et al. (2007) and Mc-Connaughey & McRoy (1979).

Response: - Our interest in these two formulas comes essentially from how widely they are applied to benthic invertebrates. For uses of McConnaughey & McRoy see for example Coat et al. 2009. This formula is also recommended in Jardine et al. (2003) a guide for stable isotopes in aquatic systems. The formula according to Post et al. (2007) is probably the most widely used lipid normalization and the authors themselves compared their formula to McConnaughey & McRoy. This wide application of lipid normalization was also pointed out by the review of Arostegui et al. 2019, who then went on to demonstrate that lipid normalization can have a large impact on mixing models. However, their study only showed that mixing models differed between normalized data and data with lipids intact. Accordingly, we think it is quite important to take the step back and compare whether we are able to achieve mixing model results after lipid normalization that resemble the results for samples that had the lipids chemically removed. We will clarify our motivation for this aspect of the paper and expand the mixing model comparison by inclusion of Bhattacharyya Coefficient for easier comparison.

Comment: From my point of view and considering the huge work made on samples treatments, the original data provided in the study is of certain value in itself and will partially contribute to increase the knowledge on stable isotopes in marine organisms. The most interesting part of the study would have been that the authors provide mathematical corrections for the conversion between preservation methods for the objective species, which should be species-specific.

Response: - While we agree that conversion should be species-specific, we are not sure how we could confidently provide these. We mentioned this briefly in the discussion (L. 323-328) but realized that this needs more explanation. The main reason why we feel we cannot propose correction factors is because the only effect we identified

as significant was the effect of formalin preservation on $\delta13C$ in comparison to the other preservation methods. All other preservations effects were small and not significant (compare section 3.1.2) and accordingly it seems very problematic to propose a correction. Nonetheless, the mixing model for dried and frozen Limecola balthica samples differed consistently and would lead to different interpretation of the species diet. For dried L. balthica samples (across all treatments) probability distributions for pPOM and SOM did not overlap considerably (we will highlight this with Bhattacharyya Coefficients) and pPOM would be clearly identified as main diet. For frozen samples, however, there is a huge overlap between pPOM and SOM and we would assume a mixed diet. We consider this a very crucial aspect of our study as it highlights that the non-significant small changes in both stable isotopes that are frequently found in the literature can strongly bias diet estimation. We will also edit the manuscript to connect the first and second part of our analysis more clearly to point out how changes in isotope ratios translate to the mixing models.

Comment: About the second part of the study, there are several constrains. The first one refers to the values applied to trophic enrichment factors (TEF) for d13C and d15N. Generalist TEFs are useful when we do not know the actual TEF for a given species, which is the case of this study. However, in some cases those TEFs are unrealistic for a given species, e.g. Planas et al. (2020). (Effect of diet on breeders and inheritance in syngnathids: application of isotopic experimentally derived data to field studies. MEPS https://doi.org/10.3354/meps13315).

Response: - We agree that TEF have to be chosen with care and should be chosen as species-specific, stage-specific, and tissue-specific as the literature allows. Unfortunately, this has not been very clear in our manuscript, but we have selected the TEF as consumer-prey pair specific as possible and we will include a justification for each of them in the method section. As mentioned above, we have re-evaluated the TEFs we chose and have come to the conclusion to adjust the TEF for $\delta13C$ for Crangon crangon according to Antonio et al. (2011) who studied Crangon uritai and its

prey. We believe we have chosen adequate TEF for the chosen species (or at least as good as possible). As you correctly pointed out, we cannot know whether these TEFs are correct or not (This is impossible in all uncontrolled field studies). However, we argue that even if they should be somehow wrong, they are equally wrong for all models for each species in our study and accordingly the effect of preservation and pre-treatment is real. We will however discuss the issue of choosing TEF and how our data compares to the TEFs we used. Furthermore, the uncertainty in TEF is accounted for within the MixSIAR framework if an error term that includes residual error is used (Stock and Semmens 2016). According to Stock and Semmens it is essential that mixing model users are faithfully incorporate uncertainty in trophic discrimination factors and expressed that they suspect mixing model users tend to underestimate TDF variance when using borrowed values. We believe we have chosen appropriately large TEF variance considering the uncertainty about our TEFs. Another thing we feel the need to point out in this context is that our study objects are lower trophic level benthic invertebrates. While we are aware of the progress that has been made for vertebrates with regard to TEF, tissue specific lipid correction, and several other methodological uncertainties, this knowledge is not available for the vast majority of benthic invertebrates (and probably never will be). Accordingly, there is no other option than applying rather generalist TEF and also generalist lipid normalization or other mathematical correction in the study of benthic food webs.

Comment: Hence, the use of unspecific TEF values should not be used as reference for checking "improvement" of mixing model outputs. The use of the term "improvement" is another issue in this study. As the authors claim, an objective of the mixing models analysis performed was to assess whether normalization and mathematical corrections should be used to adjust data for the use in such models and to improve modelling results. Achieving an improved model imply a comparison with a control model, which is lacking in the study. The use of generalist TFEs do not ensure that model outputs are actual references for comparisons.

Response: - We will avoid the use of the word "improved". However, we think we are able to evaluate whether the correction achieved what we want it to achieve due to the large variety of treatments applied to our samples. For example: We have probability distributions for (a) samples with lipids intact, (b) samples treated the same way as (a) but lipids removed, and (c) data for samples from (a) but lipid normalized. Accordingly, we can assess whether the probability distribution for (c) is closer to (b) than to (a). We will also use the Bhattacharyya Coefficient in this context to make this comparison easier to follow. Since we use the same TEF across all models, this comparison can be made. The chosen TEF might be good or bad, but it is equal across all models.

Comment: Besides, mixing models provide relative contribution estimates of dietary sources. Ideally, overlapping significance of estimates should be analyzed (Bhattacharyya's coefficients) to assess the accuracy of source contribution estimates (Bhattacharyya, A., 1943. On a measure of divergence between two statistical populations defined by their probability distributions. Bull Calcutta Math Soc 35:99-109).

Response: - We will calculate BC for pairwise comparisons and use the commonly applied BC > 0.6 to infer significant overlap between distributions.

Comment: Mixing models were carried out using a "long" run approach (which is not always sufficiently long as expected). Even though diagnostics (Gelman-Rubin and Geweke) might be fine with a long run, important differences in outputs might arise sometimes when the model is submitted to an "extreme" run. Did the authors assayed any extreme run? If so, were there any differences compared to a long run?

Response: - We did some extreme runs initially and there were no differences to the long runs. However, to avoid raising any doubt regarding the length of model runs we decided to run all models on extreme runs. This is ongoing (approximately 40% of model runs are done) and until now we have not observed differences. But we will adjust the manuscript if necessary

Comment: Finally, the use of isotopic "depletion" or "enrichment" is not correct when referring to values obtained from sample treatments. The terms "decrease" or "increase" must be used instead.

Response: - We will edit the manuscript accordingly

References:

Antonio ES, Akihide Kasai, Masahiro Ueno, Yuka Ishihi, Hisashi Yokoyama, Yoh Yamashita, Diet Shift in the Sand Shrimp Crangon Uritai along the Estuary-Marine Gradient, Journal of Crustacean Biology, Volume 31, Issue 4, 1 October 2011, Pages 635–646, https://doi.org/10.1651/10-3424.1

Arostegui, M.C., Schindler, D.E. & Holtgrieve, G.W. Does lipid-correction introduce biases into isotopic mixing models? Implications for diet reconstruction studies. Oecologia 191, 745–755 (2019). https://doi.org/10.1007/s00442-019-04525-7

COAT, S., MONTI, D., BOUCHON, C. and LEPOINT, G. (2009), Trophic relationships in a tropical stream food web assessed by stable isotope analysis. Freshwater Biology, 54: 1028-1041. doi:10.1111/j.1365-2427.2008.02149.x

Hopkins JB III, Ferguson JM (2012) Estimating the Diets of Animals Using Stable Isotopes and a Comprehensive Bayesian Mixing Model. PLoS ONE 7(1): e28478. https://doi.org/10.1371/journal.pone.0028478

Jardine, T.D., S.A. McGeachy, C.M. Paton, M. Savoie, and R.A. Cunjak. 2003. Stable isotopes in aquatic systems: Sample preparation, analysis, and interpretation. Can. Manuscr. Rep. Fish. Aquat. Sci. No. 2656: 39 p.

Parnell, A.C., Phillips, D.L., Bearhop, S., Semmens, B.X., Ward, E.J., Moore, J.W., Jackson, A.L., Grey, J., Kelly, D.J. and Inger, R. (2013), Bayesian stable isotope mixing models. Environmetrics, 24: 387-399. doi:10.1002/env.2221

Post, D.M., Layman, C.A., Arrington, D.A. et al. Getting to the fat of the matter: models, methods and assumptions for dealing with lipids in stable isotope analyses. Oecologia

152, 179–189 (2007). https://doi.org/10.1007/s00442-006-0630-x

Stock BC, Jackson AL, Ward EJ, Parnell AC, Phillips DL, Semmens BX. 2018. Analyzing mixing systems using a new generation of Bayesian tracer mixing models. PeerJ 6:e5096 https://doi.org/10.7717/peerj.5096

Stock, B.C. and Semmens, B.X. (2016), Unifying error structures in commonly used biotracer mixing models. Ecology, 97: 2562-2569. doi:10.1002/ecy.1517

---

## Referee Comment (RC3) · Anonymous Referee #2 · 13 Jul 2020

The authors of this study are making efforts to adapt the original manuscript to the reviewers' comments, especially regarding Mixing models. The responses provided to my comments on the former review comments seem to be satisfactory. For that reason, I look forward to review a much-improved and refocused new version of the manuscript.